# Asymptomatic bacteriuria among pregnant women attending antenatal care at Mbale Hospital, Eastern Uganda

**Julius Nteziyaremye[1,2], Stanley Jacob Iramiot[3], Rebecca Nekaka[4], Milton W. Musaba[2], Julius Wandabwa[2], Enoch Kisegerwa[5], Paul Kiondo[1] ***

**1** Department of Obstetrics and Gynecology, College of Health Sciences, Makerere University, Kampala, Uganda, **2** Department of Obstetrics and Gynecology, Faculty of Health Sciences, Busitema University, Mbale, Uganda, **3** Department of Microbiology and Immunology, Faculty of Health Sciences, Busitema University, Mbale, Uganda, **4** Department of Community and Public Health, Faculty of Health Sciences, Busitema University, Mbale, Uganda, **5** Department of Obstetrics and Gynecology, Mulago Hospital, Kampala, Uganda

* kiondop@yahoo.com

## Abstract

### Background

Asymptomatic bacteriuria in pregnancy (ASBP) is associated with adverse pregnancy outcomes such as pyelonephritis, preterm or low birth weight delivery if untreated. The aim of this study was to determine the prevalence of asymptomatic bacteriuria, the isolated bacterial agents, and their antibiotic sensitivity patterns in pregnant women attending antenatal care at Mbale Hospital.

### Methods

This was a cross sectional study in which 587 pregnant women with no symptoms and signs of urinary tract infection were recruited from January to March 2019. Mid-stream clean catch urine samples were collected from the women using sterile containers. The urine samples were cultured using standard laboratory methods. The bacterial colonies were identified and antibiotic sensitivity was done using disc diffusion method. Chi squared tests and logistic regression were done to identify factors associated with asymptomatic bacteriuria. A p value < 0.05 was considered statistically significant.

### Results

Out of the 587 pregnant women, 22 (3.75%) tested positive for asymptomatic bacteriuria. Women aged 20–24 years were less likely to have ASBP when compared to women aged less than 20 years (AOR = 0.14, 95%CI 0.02–0.95, *P = 0.004*). The most common isolates in descending order were *E. coli* (n = 13, 46.4%) and *S.aureus* (n = 9, 32.1%). Among the gram negative isolates, the highest sensitivity was to gentamycin (82.4%) and imipenem (82.4%). The gram positive isolates were sensitive to gentamycin (90.9%) followed by imipenem (81.8%). All the isolates were resistant to sulphamethoxazole with trimethoprim

**Data Availability Statement:** All the relevant data are within the manuscript and its supporting information files.

**Funding:** The funders had no role in the study design, data collection and analysis, decision to publish, or preparation of the manuscript.

**Competing interests:** The authors have declared that no competing interests exist

(100%). Multidrug resistance was 82.4% among gram negative isolates and 72.4% among the gram positive isolates.

## Conclusion

There was high resistance to the most commonly used antibiotics. There is need to do urine culture and sensitivity from women with ASBP so as to reduce the associated complications.

## Introduction

Urinary tract infection is a common bacterial infection in women because of the short urethra which can easily be contaminated with microorganisms from the gastrointestinal tract[1]. Pregnant women are at an increased risk of urinary tract infection because of anatomic and physiological changes of pregnancy that give a conducive environment for bacterial proliferation. Under the influence of progesterone, there is smooth muscle relaxation, dilatation of the ureters and renal pelvis especially the right due to compression from the enlarging dextrorotated uterus. In addition to the relative stasis of the urine due to reduced peristalsis of the ureters, there is glycosuria of pregnancy and general decline in the immunity [2].

Women with urinary tract infection may present with symptoms or may remain asymptomatic. Asymptomatic bacteriuria in pregnancy (ASBP) is defined as presence of bacteria in urine of quantitative counts of $10^5$ colony forming units/mL without signs and symptoms of urinary tract infection [1]. Globally the prevalence of ASBP is estimated to be 2–11%, although higher rates have been reported in Uganda [3, 4]. Women at increased risk of ASBP include women with diabetes mellitus and gestational diabetes, women of low socioeconomic status and past history of urinary tract infection[5]. Women with ASBP are at an increased risk of adverse maternal outcomes such as 30–40% incidence of pyelonephritis and this will lead to adverse fetal outcomes like premature birth and low birth weight [6].

Treatment of ASBP prevents pyelonephritis and reduces the risk of preterm deliveries [7]. Many authorities have adopted routine screening and treatment for ASBP as part of antenatal care guidelines. There is a debate on whether treatment of ASBP improves neonatal outcomes and whether antibiotic treatment is associated with adverse pregnancy outcomes. However, there is insufficient evidence to support these associations [8, 9]. It is important therefore to screen pregnant women and offer treatment to mothers diagnosed with ASBP. This will prevent later development of obstetric complications [10, 11].

In Uganda, previous studies reported prevalence of asymptomatic bacteriuria among pregnant women to range from 12.2%-13.1% [3]. *E. coli.*, *Staphylococcus epididymis*, *Staphylococcus aureus*, *and Klebssiela pnuemoniae* were the most common bacteria isolated from women with ASBP [4,12]. However, in Uganda like many other low and middle income countries, screening for ASBP is not done routinely during antenatal care. Little is known about the burden, bacterial aetiology and, sensitivity pattern of ASBP in women in Eastern Uganda. Moreover, the emergence of antimicrobial drug resistance by most uropathogens presents a challenge to the treatment of the women affected [13]. This is further complicated by the surge in the multidrug resistant organisms which do not respond to the most commonly used antibiotics [14].

The purpose of this study was to establish the prevalence of asymptomatic bacteriuria in pregnancy and the sensitivity patterns of the isolated uropathogens in women attending antenatal care at Mbale Hospital in Eastern Uganda.

## Methods

### Study design

We carried out a cross-sectional study at Mbale Regional Referral Hospital from January to March 2019.

### Setting

This study was conducted in Mbale Hospital. Mbale Hospital is a regional referral hospital in Eastern Uganda and a teaching hospital for Busitema University Faculty of Health Sciences. This is government run, charge free tertiary level hospital with a catchment population of about 4 million. On average 600 mothers attend antenatal clinics every month and delivers about 200 mothers per month.

### Study participants

Participants in this study were pregnant women who had come to attend antenatal clinic at Mbale Regional Referral Hospital.

**Eligibility criteria.** Participants included in this study were pregnant women aged 15–49 years who had come to attend antenatal clinic at Mbale Hospital. Women were excluded if they had signs and symptoms suggestive of urinary tract infection, they had vaginal bleeding or if they had used antibiotics in the previous two weeks by the time they came to the antenatal clinic.

### Sample size calculation

The sample size was calculated using a formula for comparing two proportions[15]. Using a proportion 23% for age of 35 years and above and, 9.6% for parity of five or above which were the factors associated with asymptomatic bacteriuria as was found in a study by Mayanja et.al [16], a sample size of 587 women was sufficient with power of 80%, confidence level of 95% in order to detect an odds ratio of at least 2.

### Selection of participants

We used systematic sampling to select participants for inclusion in the study. On a daily basis, women who come to attend antenatal clinic are registered on arrival as part of the routine in the clinic. Using the antenatal clinic register, the starting point (participant) was randomly selected by simple random sampling. Every third mother was selected for screening to be included in the study. In case the selected mother was found to be ineligible for inclusion in the study (failed screening), she was replaced by the next selection until the entire sample size was achieved. The women were checked for eligibility by the research as described above. The selected women were given information about the ongoing study by the research assistants on the day of recruitment during the health education sessions. The women who accepted to join the study were taken through an informed consent process in order to obtain informed consent.

### Data collection procedures

The women's sociodemographic characteristics, medical factors were collected at recruitment using an interviewer administered questionnaire. Sociodemographic characteristics included information on age in years, marital status, educational level and the socioeconomic status. Socioeconomic status was assessed using occupational status. Medical factors included history

of medical diseases and obstetric history. Mid-stream clean catch urine samples were collected by the women after being instructed on how to collect a midstream clean catch sample using sterile containers and transported to the laboratory for analysis within 2hrs.

**Bacterial isolation and antibiotic susceptibility testing.** Urinalysis was done using leucocytes esterase/nitrite urine dipsticks. Positive samples for leucocytes were sent for culture and sensitivity. Briefly, 100μml of urine was inoculated on CLED, MacConkey and chocolate agar plates. The plates were incubated at 37˚C for 18-24hrs. A diagnosis of ASBP was made when there were bacterial counts of $\geq 10^5$ Colony forming units (CFU)/ml of urine. Counts below $10^4$Cfu/ml were considered as contamination and further tests would not be performed unless the organisms were *Enterobacteriaceae*. Bacterial identification was done using colony morphology on culture plates, microscopic appearance on Gram stain and biochemical tests using standard laboratory methods of identification. The susceptibility of the isolates was determined using the Kirby-Bauer disc diffusion and dilution methods against Imipenem, Ciprofloxacin, Amoxycillin/clavulanic acid, sulfamethoxazole-trimethoprim, Cefotaxime, Ceftazidime, Gentamycin, Clindamycin, Erythromycin, Penicillin and Nitrofurantoin as recommended by the clinical laboratory standards institute [17]. Ten percent of the samples were taken to Busitema University Microbiology laboratory and HPD diagnostic laboratories for quality control purposes.

**Phenotypic detection of resistance mechanisms.**

*Determination of Methicillin Resistant Staphylococcus aureus*. Methicillin Resistance was evaluated using cefoxitin disc (10μg) on Mueller-Hinton agar (Oxoid) plate containing 2% NaCl. An inhibition zone diameter of ≤ 21 mm indicated Methicillin Resistant *Staphylococcus aureus* (MRSA).

*Screening for potential ESBL-producing isolate*. Identification of a potential ESBL-producing isolate was done using Ceftazidime disc (30μg) and/or Cefotaxime disc (30μg). An inhibition zone size of ≤ 22mm with Ceftazidime (30μg) and / or ≤ 27mm with Cefotaxime (30μg) indicated a potential ESBL producer and selected for confirmation using combination disk test (CDT) as recommended by clinical and laboratory standards institute (CLSI) guidelines[17].

Detection of extended spectrum β-lactamases was performed using a combined disc test. A disk of Ceftazidime (30μg), Cefotaxime (30μg) and Ceftazidime + Clavulanic acid (30μg/10μg), Cefotaxime+ Clavulanic acid (30μg/10μg), were placed at appropriate distance (15mm apart) on a Muller-Hinton Agar (MHA) plate. A bacterial suspension equivalent to 0.5 McFarland turbidity standards was inoculated and incubated overnight (18-24hrs) at 37˚C. An increase in the inhibition zone diameter of greater than 5 mm for a combination disc versus ceftazidime or cefotaxime disc alone was an indication of ESBLs production.

*Detection of inducible clindamycin resistant Staphylococcus aureus*. Inducible clindamycin resistance was determined using the D-test. The D-test was performed on isolates that were resistant to erythromycin but sensitive to clindamycin by placing both clindamycin and erythromycin discs 15 mm apart from the center of the Mueller-Hinton agar plate and incubated for 18-24hrs at 37˚C. Flattening on the side of erythromycin was read as inducible clindamycin resistance while a zone of clearance towards the side of erythromycin was read as clindamycin sensitive [18]

## Data management and analysis

Data collected was double entered and cleaned using EpiData 3.1 and imported to STATA version 15 for analysis. The prevalence of ASBP was computed as proportion by dividing the number of positive cultures with the total sample size and reported as a percentage. The Chi $(X^2)$ squared test and Fischer's exact test were used to find the association between maternal sociodemographic, medical and obstetric factors with ASBP. All factors found significant at

bivariate analysis (P-Value <0.05) were entered in a stepwise multivariable logistic regression model to find the factors that were independently associated with ASBP and results are presented as adjusted odds ratios with corresponding 95% confidence intervals.

**Multiple antibiotic resistance indices (MARI).** The MARI was calculated by dividing the number of antibiotics to which the microorganism was resistant by the number of antibiotics the organism was tested for sensitivity.

## Ethical considerations

Ethical approval was obtained from Makerere University School of Medicine Research and Ethics committee (SOMREC) (Ref: # REC REF 2018–186), Mbale Regional Referral Hospital Research and Ethics Committee (Ref: MRRH-REC OUT-COM/AD 02/2019) and the Uganda National Council for Science and Technology. All participants gave written informed consent. All laboratory results were availed to the participating clinicians for the management of the women.

## Results

The socio demographic characteristics of the five hundred eighty seven participants are shown in Table 1.

A total of 587 pregnant asymptomatic women were enrolled in this study. Most mothers had secondary education or higher, were married, were multigravida and were HIV negative. Women age 20–24 years were less likely to have ASBP when compared to women aged 20 years and below (Table 1).

On multivariable logistic regression, the odds of ASBP were 0.14 times lower in the 20-24yrs age group compared to those below 20yrs of age (AOR = 0.14, 95%CI 0.02–0.95, *P = 0.044*). (Table 2)

### Prevalence of ASBP

Twenty-two (3.75%) of the 587 participants recruited, had asymptomatic bacteriuria. Twenty-eight organisms were isolated from 22 positive urine cultures because some women had dual infection; The most prevalent organism isolated was *E. coli* at 46.4% (13/28), followed by *Staphylococcus aureus* at 32.1%(9/28); 53.5% of all the isolates (n = 17/28) belonged to the Enterobacteriaceae. (Table 3).

### Antimicrobial susceptibility of the isolates

Overall, the rate of sensitivity of the gram negative organisms to gentamycin, imipenem, ciprofloxacin, cefotaxime, ceftazidime, amoxicillin with clavulanic acid and nitrofurantoin were 82.4%, 82.4%, 76.5%, 64.7%, 64.7%, 19.4% and 9.4% respectively (Table 4). All the gram negative isolates were resistant to sulphamethoxazole-trimethoprim. *E. coli* showed the highest sensitivity to imipenem (92.3%) followed by ciprofloxacin (84.6%) and gentamycin (84.6%), ceftazidime (76.9%) and cefotaxime (76.9%). In addition, the sensitivity level to amoxicillin with clavulanic acid was (38.5%) and nitrofurantoin (38.5%). All the organisms were resistant to sulfamethoxazole with trimethoprim (100%).

Overall rate of sensitivity to the gram positive organisms was to imipenem, gentamycin, ciprofloxacin, Clindamycin, amoxicillin with clavulanic acid, sulphamethoxazole-trimethoprim and erythromycin were 90.9%, 81.8%, 72.7%, 63.6%, 45.5%, 37.5%, 27.3% respectively (Table 4). All gram positive organisms were resistant to penicillin. All the *S. aureus* organisms were sensitive to imipenem (100%) and gentamycin (100%), ciprofloxacin (77.8%) and

**Table 1. Socio demographic and clinical factors the 587 participants who attended antenatal clinic at Mbale Regional Referral Hospital.**

| Variable | Positive (n = 22) | Negative(n = 565) | Crude OR | 95% CI | P value |
|---|---|---|---|---|---|
| **Age(yrs)** | | | | | |
| Median(IQR) | 28.5(6) | 26(7) | | | |
| <20 | 3(13.64) | 37(6.55) | 1 | | |
| 20–24 | 2(9.09) | 190(33.63) | 0.13 | 0.021–0.804 | 0.028 |
| 25–29 | 8(36.36) | 174(30.80) | 0.57 | 0.14–2.23 | 0.418 |
| >29 | 9(40.91) | 164(29.03) | 0.68 | 0.17–2.6 | 0.572 |
| **Education level** | | | | | |
| Secondary and below | 13(61.9) | 425(75.4) | 1.5 | 0.8–2.7 | 0.221 |
| Tertiary | 8(38.10) | 139(24.65) | | | |
| **Married** | | | | | |
| No | 3(14.29) | 49(8.78) | 1 | | |
| Yes | 18(85.71) | 509(91.22) | 1.7 | 0.5–6.1 | 0.392 |
| Gestational age (weeks) | | | | | |
| Median (IQR) | 29.75(12.9) | 26.6(13) | 1.03 | 0.98–1.08 | 0.266 |
| **Parity** | | | | | |
| Primegravida | 5(22.7) | 156(28.1) | 1 | | |
| Multigravida | 17(77.3) | 399(71.9) | 1.3 | 0.5–3.7 | 0.582 |
| **HIV status** | | | | | |
| Negative | 20(90.91) | 504(89.52) | 1 | | |
| Positive | 2(9.09) | 59(10.48) | 0.85 | 0.19–3.7 | 0.835 |
| **Occupation** | | | | | |
| Informal | 13(61.9) | 390(71.69) | 1 | | |
| Formal | 8(38.1) | 154(28.31) | 1.6 | 0.63–3.8 | 0.334 |
| Random blood sugar(mmol/l) | | | | | |
| Median (IQR) | 1.89±5.4 | 1.38±5 | 1.1 | 0.9–1.3 | 0.568 |
| MUAC(cm) | | | | | |
| Mean(SD) | 28.6±3.9 | 28.1±3.9 | 1.03 | 0.93–1.14 | 0.557 |

IQR: Interquartile range, OR: Odds ratio, CI: Confidence interval, MUAC: Mid upper arm circumference.

clindamycin (77.8%). In addition, the sensitivity to amoxicillin with cavulanic acid, erythromycin and sulfamethoxazole-trimethoprim was 33.3%, 33.3% and 11.1% respectively. All the *S. aureus* organisms were resistant to penicillin.

**Table 2. Multivariable analysis of factors associated with ASBP in 587 participants attending antenatal clinic at Mbale Regional Referral Hospital.**

| Variable | Crude OR | Adjusted OR | 95% CI | P value |
|---|---|---|---|---|
| **Age** | | | | |
| <20 | 1 | 1 | | |
| 20–24 | 0.13 | 0.14 | 0.02–0.95 | **0.044** |
| 25–29 | 0.6 | 0.5 | 0.12–2.8 | 0.491 |
| >29 | 0.7 | 0.67 | 0.14–4.2 | 0.76 |
| **Gravidity** | | | | |
| Prime gravida | | 1 | | |
| Multigravida | | 0.99 | 0.95–1.06 | 0.989 |
| **Gestational age (weeks)** | 1.03 | 1.001 | 0.95–1.06 | 0.96 |

**Table 3. Bacterial isolates from urine samples of pregnant women attending antenatal clinic at Mbale Hospital.**

| Type of bacterial isolate | Number (%) |
|---|---|
| **Gram negative isolates** | |
| *E.coli* | 13 (46.4) |
| *Klebsiella pneumoniae* | 2 (7.1) |
| *Pseudomonas auregenosa* | 2 (7.1) |
| **Gram positive organisms** | |
| *Staphylococcus aureus* | 9 (32.1) |
| *Enterococcus* | 2 (7.1) |
| **Total** | 28(100) |

Overall, 82.4% (14/17) of the gram negative isolates were multidrug resistant (MDR). The highest MDR level was seen among the *Pseudomonas* (100%, 2/2), *Klebsiella* (100%, 2/2) and *E. coli* (92.3%, 12/13). The overall MDR level among the gram positive isolates was 72.7% (n = 8/11). The highest MDR level was observed among *Enteroccocus spp* (100%, 2/2) followed by *S. aureus* at (66.7%, 6/9) MDR level (Table 5).

The phenotypic mechanisms of resistance by the gram negative isolates were extended beta-lactamase production: *E. coli* (30.7%), *Klebsiella pneuminiae* (50%) and Pseudomonas aurogenosa (100%). Among the gram positive isolates, Staphylococcus aureus exhibited Methicillin resistant staphylococcus aureus (MRSA) 60% and Inducible Clindamycin resistance, 40%.

The results of multiple antibiotic resistance indices (MARI) of the bacterial isolates are shown in Table 6.

## Discussion

Pregnant women are at an increased risk of acquiring urinary tract infection due to functional and anatomical changes in pregnancy. In most cases the urinary tract infection is asymptomatic. The prevalence of asymptomatic urinary tract infection in women attending antenatal in Mbale Hospital was 3.75%. This is similar to what was found in a study in Egypt [19] and in a study in Turkey[20]. However, it is lower than what was reported in other studies in Uganda [3, 21] and elsewhere [6, 14, 22]. Earlier studies in Uganda were carried in an urban setting. The women studied earlier were of low socioeconomic status and stayed in slums with poor hygiene all of which increase the risk of developing urinary tract infection [23]. It is also possible that some women were using antibiotics at the time of sample collection which could explain the low prevalence in our study as there are a number of privately owned health facilities that have made access to antibiotics easier. In fact all the microorganisms isolated in this study showed a MARI of more than 0.20. MARI values of greater than 0.20 suggest that the strains of such bacteria are from an environment where there is overuse or indiscriminate use of antibiotics[24]. This suggests that a big proportion of the isolates have been exposed to many antibiotics and have developed resistance to these antibiotics.

If asymptomatic urinary tract infection in pregnancy is not treated, it is associated with 30% risk of developing pyelonephritis [25] with subsequent low birth weight and/or preterm delivery. Asymptomatic urianry tract infection in pregnancy is common because the short urethra in women makes the urinary tract to be easily contaminated with fecal flora. [26]

In this study, *E. coli* was the most common organism isolated. This is in agreement with other studies carried in Uganda[3] and elsewhere[22] in which E.coli was the most common organism isolated. *E. coli* is a common microorganism in the perineum and failure to maintain

**Table 4. The antibiotics sensitivity pattern of profile of bacterial isolates of women attending antenatal clinic at Mbale Hospital.**

| Bacterial isolates | No. of strains sensitive to antibiotics (%) | | | | | | | | |
|---|---|---|---|---|---|---|---|---|---|
| Gram negative isolates | No. | IPM | CIP | AMC | F | SXT | CTX | CAZ | CN |
| *E.coli* | 13 | 12(92.3) | 11(84.6) | 5(38.5) | 5(38.5) | 0(0) | 10(76.9) | 10(76.9) | 11(84.6) |
| *Pseudomonas Aerugenosa* | 2 | 1(100) | 1(50) | 0(0) | 0(0) | 0(0) | 0(0) | 0(0) | 2(100) |
| *Klebsiella Pneumoniae* | 2 | 2(100) | 1(50) | 0(0) | 0(0) | 0(0) | 1(50) | 1(50) | 1(50) |
| **Total** | 17 | 14(82.4) | 13(76.5) | 5(19.4) | 5(19.4) | 0(0) | 11(64.9) | 11(64.9) | 14(82.4) |
| Gram positive isolates | No. | IPM | CIP | AMC | PEN | SXT | DA | E | CN |
| *Staphylococcus aureus* | 9 | 9(100) | 7(77.8) | 3(33.3)) | 0(0) | 1(11.1) | 7(77.8) | 3(33.3) | 9(100) |
| *Enterococcus* | 2 | 1(50) | 1(50) | 2(100) | 0(0) | 2(100) | 0(0) | 0(0) | 0(0) |
| **Total** | 11 | 10(90.9) | 8(72.7) | 5(54.5) | 0(0) | 3(37.5) | 7(63.5) | 3(27.3) | 9(81.8) |

**Key**: Imipenem(IPM) **10μg**, Ciprofloxacin(CIP) **30μg**, Amoxicillin with clavulanic acid(AMC) **30μg**, Nitrofurantoin(F) **30μg**, Sulfamethoxazole-trimethoprim(SXT) **25μg**, Cefotaxime(CTX) **30μg**, Ceftazidime(CAZ) **30μg**, Gentamycin(CN)**10μg**, Penicillin(P) **10μg**, Clindamycin(DA) **2μg**, Erythromycin(E) **10μg**.

personal hygiene may increase the risk infection with *E. coli*. [26]. In addition, gram negative bacteria have a distinct structure which enables the organism to attach, grow and invade the uro-epithelium. This may result in invasive infection and pyelonephritis [25].

The second most common isolated organism was *Staphylococcus aureus*. This is agreement with other studies [4, 14] in which *Staphylococcus aureus* was the second most commonly isolated Uropathogen. However, it disagrees with other studies [27]in which Staphylococci was the most commonly isolated organism. The presence of staphylococcus in the urine is due poor genital hygiene by the women.

In this study, the gram negative isolates were resistant to commonly used antibiotics. *E. coli*, the most common isolate was sensitive to gentamycin and imipenem. This is similar to what has been found in other studies [3, 14, 28] in which the gram negative isolates are resistant to commonly used antibiotics.

Similarly, we found that the gram positive isolates were resistant to commonly used antibiotics (amoxicillin with clavulanic acid, sulphamethoxazole with trimethoprim, erythromycin, and penicillin). *Staphylococcus aureus* was sensitive to imipenem and gentamycin. This is in line with what has been found in other studies [4, 29]but differs with what was found in a study from Turkey [30]. The resistance to the commonly used antibiotics could be due overuse or misuse of these antibiotics [31].

All the isolates were found to have a high level multidrug resistance (MDR). This is similar to what was found in studies in Uganda [32, 33], Ethiopia [34] and Nepal [35]. Strains isolated in this study were Methicillin-Resistant Staphylococcus aureus (MRSA) and Inducible Clindamycin resistance among the gram positive isolates and Expanded Spectrum beta lactam (ESBL) producing organisms among the gram negative isolates. The emergence of MDR has been associated with wide spread use of antibiotics, inappropriate drug use and weak antibiotic monitoring. This leads to selection of antibiotic resistance mechanisms in bacteria. The emergence of MDR isolates is a serious problem. It compromises the activity of the broad spectrum antibiotics and is a major therapeutic challenge with great impact on patient outcomes. The

**Table 5. Multidrug resistance of the uropathogens.**

| Isolate | *E.coli* | *Pseudomonas aeruginosa* | *Klebsiella pneumonia* | *Staphylococcus aureus* | *Enterococcus feacum* |
|---|---|---|---|---|---|
| Level of resistance to three drugs or more | 12(92.3%) | 2(100%) | 2(100%) | 6(66.7%) | 2(100%) |
| Overall Multidrug resistance | Gram negative organisms 14(82.4%) | | | Gram positive organisms 8(72.7%) | |

**Table 6. Multiple antibiotic indices (MARI) of the bacterial isolates.**

| Isolates | MARI | Antibiotics to which the organisms are resistant |
|---|---|---|
| *E. coli* | 1.0 | IPM, CIP, AMC, F, SXT,CTX, CAZ and CN |
| *P. aeruginosa* | 0.75 | CIP, AMC, F, SXT,CTX and CAZ |
| *K. pneumoniae* | 0.88 | CIP, AMC, F, SXT,CTX, CAZ and CN |
| *S. aureus* | 0.75 | CIP, AMC, PEN, SXT, DA and E |
| *E. faecum* | 0.75 | IPM, CIP, PEN, DA, E and CN |

Total number of antibiotics tested = 8

IPM: Imipenem, CIP: Ciprofloxacin, AMC: Amoxacillin with Clavulanic acid, F: Nitrofurantoin, CXT: Ceftazidime,
CN: Gentamycin, PEN: Penicillin, DA: Clindamycin, E: Erythromycin

From the MARI obtained in this study all the bacterial isolates gave a MARI of >0.20.

resistance to β lactam antibiotics is due to production of Extended Spectrum β Lactamase production by the gram negative isolates, acquisition of resistance genes, down regulation of receptors and drug efflux[28, 33, 35]. This is mainly due to selective pressure generated by the use of the third generation cephalosporins [36]. This creates a major challenge to National guidelines as the organisms were resistant to most commonly used drugs leaving the less commonly used drugs and expensive drugs.

## Conclusion

The resistance of the isolates to commonly used antibiotics was high. There is need to do urine culture and sensitivity from women diagnosed with asymptomatic bacteriuria in pregnancy so that appropriate antimicrobial agents are used in order to reduce the associated complications.

## Supporting information

**S1 Dataset.**
(DTA)

## Acknowledgments

Authors would like to thank the Mbale Regional Referral Hospital antenatal clinic staff, the participants, the Busitema University Microbiology laboratory staff and the Busitema University Directorate of graduate Science and Innovation

## Author Contributions

**Conceptualization:** Julius Nteziyaremye, Milton W. Musaba, Julius Wandabwa, Enoch Kisegerwa, Paul Kiondo.

**Data curation:** Julius Nteziyaremye, Paul Kiondo.

**Formal analysis:** Julius Nteziyaremye, Stanley Jacob Iramiot, Rebecca Nekaka, Milton W. Musaba, Julius Wandabwa, Paul Kiondo.

**Funding acquisition:** Julius Nteziyaremye.

**Investigation:** Julius Nteziyaremye, Stanley Jacob Iramiot, Rebecca Nekaka, Milton W. Musaba.

**Methodology:** Paul Kiondo.

**Project administration:** Julius Nteziyaremye.

**Supervision:** Julius Nteziyaremye, Julius Wandabwa, Enoch Kisegerwa, Paul Kiondo.

**Validation:** Paul Kiondo.

**Writing – original draft:** Julius Nteziyaremye, Stanley Jacob Iramiot, Rebecca Nekaka, Milton W. Musaba, Julius Wandabwa, Enoch Kisegerwa, Paul Kiondo.

**Writing – review & editing:** Julius Nteziyaremye, Stanley Jacob Iramiot, Rebecca Nekaka, Milton W. Musaba, Julius Wandabwa, Enoch Kisegerwa, Paul Kiondo.

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
