## [Decision Letter · Decision Letter 0]

27 Nov 2019

PONE-D-19-30829

Asymptomatic bacteriuria among pregnant women attending antenatal care at Mbale hospital, Eastern Uganda.

PLOS ONE

Dear Prof Kiondo,

Thank you for submitting your manuscript to PLOS ONE. After careful consideration, we feel that it has merit but does not fully meet PLOS ONE’s publication criteria as it currently stands. Therefore, we invite you to submit a revised version of the manuscript that addresses the points raised during the review process.

Both reviewers note some interest in this manuscript but have suggested necessary revisions that need to be made. I think that writing a more substantial Introduction section is of particular importance.

In addition to the points noted by the reviewers I would like you to consider the following points:

In the "Selection of participants" section it reads as if only every third woman who had "accepted to join" the study was used in the present analysis. Please clarify as to whether this is true. If it is true, were the 2/3 women not used in the present analysis used in other studies that were running at the same time? At present this seems rather wasteful of recruits.

Do you know if any of the women had either pre-existing or gestational diabetes? Could glycosuria have influenced your findings?

In addition please provide some information about the BMI of the study participants.

In the "Data Collection Procedures" section, why were further tests performed for Enterobacteriaceae counts less than 10,000 Cfu/ml, but not for other potential bacterial infections?

In Table 1 the "Married" is a simple binary yes or no. Were any of the study participants co-habiting?

In Table 2 the p-values do not agree for the 20-24 age group in comparison to the <20 group, in the main text of the Table and the text below it.

The section title "ANTIICROBIAL SUSCEPTIBILITY OF THE ISOLATES" contains a typographical error.

We would appreciate receiving your revised manuscript by Jan 11 2020 11:59PM. To enhance the reproducibility of your results, we recommend that if applicable you deposit your laboratory protocols in protocols.io, where a protocol can be assigned its own identifier (DOI) such that it can be cited independently in the future. For instructions see: http://journals.plos.org/plosone/s/submission-guidelines#loc-laboratory-protocols

We look forward to receiving your revised manuscript.

Kind regards,

Clive J Petry, PhD

Academic Editor

PLOS ONE

1. In your Methods section, please provide additional information about the participant recruitment method and the demographic details of your participants. Please ensure you have provided sufficient details to replicate the analyses such as: a) the recruitment date range (month and year), b)  a clear description of how sample size was calculated (at the moment, it is not clear why the parameters used were adopted), and c) a clear description of the "assets and amenities" considered to determine socioeconomic status, and and explanation on why these are not included in Table 1.

3. If your supporting tables are just duplicates of your tables which are included within your manuscript, please remove these so as not to confuse readers

Reviewers' comments:

Reviewer's Responses to Questions

**Comments to the Author**

1. Is the manuscript technically sound, and do the data support the conclusions?

Reviewer #1: Partly

Reviewer #2: Yes

2. Has the statistical analysis been performed appropriately and rigorously? 

Reviewer #1: Yes

Reviewer #2: Yes

3. Have the authors made all data underlying the findings in their manuscript fully available?

Reviewer #1: Yes

Reviewer #2: Yes

4. Is the manuscript presented in an intelligible fashion and written in standard English?

Reviewer #1: Yes

Reviewer #2: Yes

5. Review Comments to the Author

Reviewer #1: -The introduction is too short and authors did not identify other literature enough on the topic with no explaining of how the study relates to previously published research in the area. Also, references are not uptodate with most of the references more than 10 years ago.

-The study is a screening one with no details on the mechanisms of resistance found in their isolates (perhaps some more details as simple gene detection by PCR would be ).

-Ethical approval is mentioned twice.

-The authors should put bacterial isolation and antimicrobial susceptibility testing under a new heading other than "data Collection".

-Fosfomycin is an important antimicrobial usually used for urinary tract infection (perhaps it would be better if the authors test for its action).

-Table one misses a footnote.

- Ref 15 is for a study carried on children, how is it comparable to the current study!

-Resistance to beta-lactam antibiotics is not only due to ESBL production as stated by the authors in the last paragraph.

Reviewer #2: Review Comments to the Author

Please use the space provided to explain your answers to the questions above. You may also include additional comments for the author, including concerns about dual publication, research ethics, or publication ethics. (Please upload your review as an attachment if it exceeds 20,000 characters) (Limit 200 to 20000 Characters)

Could you please see my attached review, I have added all my comments to the Author in an attachment.

6. PLOS authors have the option to publish the peer review history of their article (what does this mean?). If published, this will include your full peer review and any attached files.

Reviewer #1: Yes: Noha A Hassuna

Reviewer #2: No

---

## [Author Response · Author response to Decision Letter 0]

7 Feb 2020

Response to reviewers

A more substantial introduction has been written.

Selection of participants: this has been corrected.

All the women had blood sugar tests carried out on them. None of the women had preexisting or gestational diabetes mellitus.

Mid upper mid circumference was used to calculate the BMI of the women since the women did not know their pre-pregnancy weight. This information has been included in table 1.

In the "Data Collection Procedures" section, further tests were performed for Enterobacteriaceae counts less than 10,000 Cfu/ml, but not for other potential bacterial infections, because Enterobacteriaceae are rarely contaminants whereas other bacteria such as Staphylococcus are contaminants. 

In Table 1 “Married” is simple yes or no, because women who were cohabiting were grouped under married, since they only lack a legal marriage. 

In table 2, the p-values in the main text for 20-24 age group in comparison to <20 group has been corrected in the text to agree with what is in the table

The typographic error in the title "ANTIICROBIAL SUSCEPTIBILITY OF THE ISOLATES" has been corrected

The date of recruitment has been included. 

The sample size calculation: more information has been given about the proportions which were used in the sample size calculation. 

Socioeconomic status was assessed using occupational status which has been included in table 1. 

The references have been updated.

Phenotypic mechanisms of resistance have been added in the methods section

Ethical approval appears once at the moment.

Bacterial identification and susceptibility testing have been put under a new heading.

A footnote has been added on table 1

Reference 15 is about pregnant women not children.

Other mechanisms of resistance to beta-lactam antibiotics have been included.

The positive women for asymptomatic bacteriuria are reported to be 22 in abstract and in the results. But there were 28 organisms is isolated from the 22 positive samples. This is because some women had duo infection with microorganisms. 

Multiple antibiotic resistance indices (MARI) for the microorganisms in the study have been calculated. The method which was used to calculate the indices has been included in the methods section. Table 6 shows the (MARI) for the different microorganisms in the study.

---

## [Decision Letter · Decision Letter 1]

3 Mar 2020

Asymptomatic bacteriuria among pregnant women attending antenatal care at Mbale hospital, Eastern Uganda.

PONE-D-19-30829R1

Dear Dr. Kiondo,

We are pleased to inform you that your manuscript has been judged scientifically suitable for publication and will be formally accepted for publication once it complies with all outstanding technical requirements.

With kind regards,

Clive J Petry, PhD

Academic Editor

PLOS ONE

Additional Editor Comments (optional):

Reviewers' comments:

Reviewer's Responses to Questions

**Comments to the Author**

1. If the authors have adequately addressed your comments raised in a previous round of review and you feel that this manuscript is now acceptable for publication, you may indicate that here to bypass the “Comments to the Author” section, enter your conflict of interest statement in the “Confidential to Editor” section, and submit your "Accept" recommendation.

Reviewer #1: All comments have been addressed

2. Is the manuscript technically sound, and do the data support the conclusions?

Reviewer #1: Yes

3. Has the statistical analysis been performed appropriately and rigorously? 

Reviewer #1: Yes

4. Have the authors made all data underlying the findings in their manuscript fully available?

Reviewer #1: Yes

5. Is the manuscript presented in an intelligible fashion and written in standard English?

Reviewer #1: Yes

6. Review Comments to the Author

Reviewer #1: The manuscript has improved. Authors have responded to previous corrections. All responses meet formatting specifications.

7. PLOS authors have the option to publish the peer review history of their article (what does this mean?). If published, this will include your full peer review and any attached files.

Reviewer #1: Yes: Noha A Hassuna

---

## [Editor Report · Acceptance letter]

6 Mar 2020

PONE-D-19-30829R1 

Asymptomatic bacteriuria among pregnant women attending antenatal care at Mbale hospital, Eastern Uganda. 

Dear Dr. Kiondo:

I am pleased to inform you that your manuscript has been deemed suitable for publication in PLOS ONE. Congratulations! Your manuscript is now with our production department. 

With kind regards,

on behalf of

Dr. Clive J Petry 

Academic Editor

PLOS ONE